# Higher Ultra-Processed Food Consumption Is Associated with Greater High-Sensitivity C-Reactive Protein Concentration in Adults: Cross-Sectional Results from the Melbourne Collaborative Cohort Study

**DOI:** 10.3390/nu14163309

**Published:** 2022-08-12

**Authors:** Melissa M. Lane, Mojtaba Lotfaliany, Malcolm Forbes, Amy Loughman, Tetyana Rocks, Adrienne O’Neil, Priscila Machado, Felice N. Jacka, Allison Hodge, Wolfgang Marx

**Affiliations:** 1Food & Mood Centre, The Institute for Mental and Physical Health and Clinical Translation (IMPACT), School of Medicine, Deakin University, Geelong, VIC 3220, Australia; 2Mental Health, Drugs & Alcohol Service, University Hospital Geelong, Barwon Health, Geelong, VIC 3220, Australia; 3Department of Psychiatry, University of Melbourne, Parkville, VIC 3050, Australia; 4Institute for Physical Activity and Nutrition, School of Exercise and Nutrition Sciences, Deakin University, Melbourne, VIC 3125, Australia; 5Center for Epidemiological Research in Nutrition and Health, University of Sao Paulo, Sao Paulo 01246-904, Brazil; 6Centre for Adolescent Health, Murdoch Children’s Research Institute, Parkville, VIC 3052, Australia; 7College of Public Health, Medical & Veterinary Sciences, James Cook University, Townsville, QLD 4811, Australia; 8Cancer Epidemiology Division, Cancer Council Victoria, Melbourne, VIC 3004, Australia; 9Centre for Epidemiology and Biostatistics, Melbourne School of Population and Global Health, University of Melbourne, Melbourne, VIC 3010, Australia

**Keywords:** ultra-processed food, NOVA, diet, inflammation, high-sensitivity C-reactive protein, non-communicable diseases, cross-sectional

## Abstract

Background: Few studies have examined associations between ultra-processed food intake and biomarkers of inflammation, and inconsistent results have been reported in the small number of studies that do exist. As such, further investigation is required. Methods: Cross-sectional baseline data from the Melbourne Collaborative Cohort Study (MCCS) were analysed (*n* = 2018). We applied the NOVA food classification system to data from a food frequency questionnaire (FFQ) to determine ultra-processed food intake (g/day). The outcome was high-sensitivity C-reactive protein concentration (hsCRP; mg/L). We fitted unadjusted and adjusted linear regression analyses, with sociodemographic characteristics and lifestyle- and health-related behaviours as covariates. Supplementary analyses further adjusted for body mass index (kg/m^2^). Sex was assessed as a possible effect modifier. Ultra-processed food intake was modelled as 100 g increments and the magnitude of associations expressed as estimated relative change in hsCRP concentration with accompanying 95% confidence intervals (95%CIs). Results: After adjustment, every 100 g increase in ultra-processed food intake was associated with a 4.0% increase in hsCRP concentration (95%CIs: 2.1–5.9%, *p* < 0.001). Supplementary analyses showed that part of this association was independent of body mass index (estimated relative change in hsCRP: 2.5%; 95%CIs: 0.8–4.3%, *p* = 0.004). No interaction was observed between sex and ultra-processed food intake. Conclusion: Higher ultra-processed food intake was cross-sectionally associated with elevated hsCRP, which appeared to occur independent of body mass index. Future prospective and intervention studies are necessary to confirm directionality and whether the observed association is causal.

## 1. Introduction

Nutrition science has long sought to understand the effects of diet on human health. This has largely been done by classifying foods based on their nutrient composition. The impacts on health of individual macro- and micro-nutrients as well as kilojoules have typically been considered independent of different foods and food group sources [1]. Excess intakes of sugar, salt, saturated fat, and kilojoules have been previously linked with increased risk of cardiometabolic conditions [2,3,4]. Such research has been beneficial for understanding nutritional physiology and has subsequently informed dietary recommendations [5]. However, this nutrient-centric perspective does not capture the effect of complex food matrices. A food matrix is characterised as the molecular interactions between nutrient and non-nutrient components of food [6]. Indeed, emerging experimental [7] and epidemiological [8,9,10] evidence implicates the extent to which a food has been processed (or undergone food matrix alterations) as a risk factor for chronic non-communicable diseases, morbidity, and mortality.

The NOVA (name, not acronym) food classification system was recently developed to allow for the categorisation of food items based on their level of processing: from unprocessed or minimally processed food, processed culinary ingredients, processed food, to extensively processed food termed “ultra-processed” [11]. Ultra-processed foods, are defined by NOVA as industrial formulations created from compounds extracted, derived, or synthesised from food or food substrates. Ultra-processed foods also typically contain five or more ingredients including artificial food additives (e.g., colours, texturising agents, and olfactory and taste enhancers) and are commonly inexpensive, virtually imperishable, easily consumed, and highly palatable [12]. Time-series country-level sales data from 2006 to 2019 show a substantial growth in the types and quantities of ultra-processed foods sold worldwide, with projected increases to 2024 [13,14]. This suggests a transition away from non-ultra-processed food and toward a more processed global diet [13,14].

Chronic low-grade inflammation, marked by the presence of elevated inflammatory cytokines, is both a driver of chronic diseases and a characteristic of an established diseased state [15]. These diseases include cancers [16], cardiometabolic conditions [17], and mental disorders [18,19]. The shared link between chronic low-grade inflammation and diseased states exists despite the different organs and systems involved in their onset, prognosis, and morbidity [20]. Hence, better understanding and addressing possible drivers of inflammation is of significant public health interest. However, little data are available that have directly linked ultra-processed food intake to inflammation.

In the three epidemiological studies that do exist [21,22,23], inconsistent results have been observed. These included sex- and cohort-specific differences within and between studies [21,22] as well as associations of ultra-processed food with some inflammatory biomarkers but not others [23]. Importantly, each of these three studies included samples from Brazil, where the concept of avoiding ultra-processed food has received recognition in official dietary guidelines since 2014 [24], and where consumption of ultra-processed food is estimated to be lower than higher-income countries [11,14]. Thus, there is a need for further investigation of the association between ultra-processed food intake and inflammation in other regions, particularly in settings where the substitution of non-ultra-processed foods for those that are ultra-processed is increasingly common. The current study aimed to address this gap by using data from the Melbourne Collaborative Cohort Study (MCCS) to investigate cross-sectional associations between ultra-processed food intake and plasma concentrations of the inflammatory cytokine, high-sensitivity C-reactive protein (hsCRP).

## 2. Methods

A full description of methods for data collection in the MCCS was published elsewhere [25]. In brief, the MCCS is a study that aimed to assess prospective associations between diet and lifestyle and chronic non-communicable diseases [25]. Between 1990 and 1994 (baseline), 41,513 people (24,469 women) aged between 27 and 76 (99% 40–69) years were recruited from the Melbourne metropolitan area, with migrants from Southern Europe included and deliberately recruited to expand the span of diet and lifestyle exposures. At baseline, participants completed surveys and anthropometric measurements, and blood samples were collected. A case-cohort sub-study was undertaken to provide a more economical design within which to perform assays on blood samples collected at baseline (as part of the original MCCS project) and analyse associations of various molecules in the stored plasma with selected disease outcomes; cases were included in the case-cohort if they were identified as having the outcome of interest by 30 June 2002. High-sensitivity C-reactive protein as a biomarker of inflammation was measured in incident cardiovascular death cases together with a random sample (sub-cohort) of all participants in the MCCS.

The current cross-sectional study thus comprised a sample of participants from the case-cohort sub-study for whom valid baseline dietary data and plasma hsCRP measurements were available (see Figure 1). We excluded participants who had missing hsCRP data (*n* = 161), total energy intake (kJ/day) below the 1st and above the 99th percentiles (*n* = 40), or hsCRP concentration above the 99th percentile (*n* = 23). Two thousand and eighteen participants remained for analysis, including both the cardiovascular disease death group as cases (*n* = 567) and the random sample of all participants from the original MCCS project as the sub-cohort (*n* = 1451). Cardiovascular disease death cases were identified from notifications to the Victorian Registry of Births, Deaths and Marriages, and the Australian National Death Index (codes 390–459 and I00–I99 of the International Classification of Diseases, ICD-9 and ICD-10, respectively).

The current study was reported in accordance with the Strengthening the Reporting of Observational Studies in Epidemiology (STROBE) statement and checklist for cross-sectional studies [25]. The current study was prospectively registered with Open Science Framework (OSF) registry (registration DOI: 10.17605/OSF.IO/EHFXD) and was approved for exemption from ethical review in accordance with the National Statement on Ethical Conduct in Human Research (2007, updated 2018) Section 5.1.22 by the Deakin University Human Research Ethics Committee (project number: 2020-413, received 18th of November 2020). The study protocol for the original MCCS project was approved by the Cancer Council Victoria’s Human Research Ethics Committee (project number: IEC 9001, received 23rd of August 1990). Participants provided written consent to participate including researcher access to their medical records [26].

### 2.1. Exposure: Dietary Assessment

At baseline of the original MCCS project, participants attended clinics where dietary data were collected using a self-administered 121-item food frequency questionnaire (FFQ) specifically developed for use in this multiethnic cohort [27]. This FFQ was based on weighed food records in 810 Melbournians of similar demographics to the cohort [27]. For this study, all FFQ food items were classified according to the NOVA food classification system as ultra-processed foods and non-ultra-processed foods by two experts with Australian food and dietary intake knowledge. Examples of NOVA’s classification of ultra-processed food (NOVA group 4) include soft drinks, sweet or savoury packaged snacks, confectionery, packaged breads and buns, margarine, reconstituted meat products, and pre-prepared frozen or shelf-stable dishes. Examples of NOVA’s classification of non-ultra-processed food include unprocessed or minimally processed foods (NOVA group 1) such as rice and other cereals, meat, fish, milk, eggs, fruit, roots and tubers, vegetables, and nuts and seeds; processed culinary ingredients (NOVA group 2) such as sugar, plant oils, and butter; and processed foods (NOVA group 3) such as processed breads and cheese, canned fruit and fish, and salted and smoked meats. More information regarding the NOVA food classification system can be found elsewhere (1). When it was not possible to discriminate, (e.g., food items such as ‘bread’, ‘pasta or noodles’, ‘low fat cheese’, ‘yoghurt’, and ‘fruit juice’), cross-sectional data from the National Nutrition Survey 1995–1996 (data not published) and Australian National Nutrition and Physical Activity Survey (NNPAS) 2011–2012 were used for comparison and decision making [28].

As per previous research [27,29,30], the mean daily contribution of ultra-processed foods to intake of total energy (kJ) and weight (g) was calculated by transforming frequencies into grams based on sex-specific portion sizes of each food multiplied by the daily equivalent frequency. Energy was estimated based on the Nutrient Data Table for Use in Australia 1995 (NUTTAB 95). The NUTTAB 95 is a food composition database that contains information for 1800 foods and beverages available in Australia [31].

### 2.2. Outcome: Inflammatory Cytokine Assessment

Blood samples were also collected at baseline. Venous blood samples were drawn in lithium-heparin tubes, and plasma was subsequently stored at −180 °C (liquid nitrogen) until assayed as part of the case-cohort sub-study. HsCRP concentration expressed as mg/L was measured by a high-sensitivity immunonephelometric assay. The assay was a Dade Behring nephelometric assay done on a BNII Nehphelometric Analyser, Dade-Behring Diagnostics, Lane Cove, NSW, Australia.

### 2.3. Assessment of Covariates

Potential covariates were identified based on previous literature [7,21,32,33,34,35] and included in a directed acyclic graph to map hypothesised causal relationships between all relevant variables (Appendix A Appendix A). These covariates were assessed through a structured interview that was administered at baseline providing data on sociodemographic characteristics and lifestyle- and health-related factors.

In particular, the sociodemographic characteristics included: age (continuous), sex (men and women), country of birth (Australia/New Zealand, United Kingdom/Malta, Italy, and Greece), marital status (married, de facto, single, divorced, separated, and widow), highest level of education (primary school, high/technical school, and tertiary degree or diploma) and Socio-Economic Indexes for Areas (SEIFA)—Index of Relative Socio-Economic Disadvantage [36]. SEIFA scores are recorded by the Australian Bureau of Statistics and refer to the relative socioeconomic advantage and disadvantage of defined geographical areas such as postal code [37] (we divided these scores into quintiles, with the lowest and highest representing the greatest and least disadvantaged, respectively).

The lifestyle- and health-related factors included: smoking status (never smoked, current smoker, and former smoker), leisure time physical activity over the last 6 months (a score was calculated ranging from 0–16 based on the frequency of walking and less vigorous and vigorous activity multiplied by two, which was then divided into categories, namely: 0 (none), >0 and <4 (low), ≥4 and <6 (moderate), and ≥6 (high) [32,34]), and alcohol intake using beverage-specific quantity frequency questions (lifetime abstainers, ex-drinkers, and current drinkers (further categorised as up to 19, 20–29, 30–39, and 40+ g/day)) [26,32,33]. Height and weight were measured, and body mass index was calculated as kg/m^2^ [26]. These sociodemographic characteristics and lifestyle- and health-related variables were used as covariates.

### 2.4. Statistical Analyses

An inverse probability weighting method was applied to address the case-cohort design and adjust for the possibility of oversampling cases versus participants from the sub-cohort [26]. Characteristics of participants were summarised using mean and standard deviation (SD) or median and interquartile range (IQR) for continuous variables and frequency and percentage for categorical variables.

Linear regression analysis was used to examine associations between the consumption of ultra-processed food and hsCRP concentration. We aimed to better account for ultra-processed formulations that did not provide energy or were low in energy (e.g., artificially sweetened beverages) as per [38,39,40]. Thus, the total weight of ultra-processed foods in grams per day (g/day) was adjusted for energy using Willet’s residual method [41] and used to model our exposure in 100 g increments. We chose 100 g increments to aid in reporting and interpretation. We assessed our outcome variable, hsCRP concentration, continuously; however, because the variance of residuals for hsCRP concentration was not homogeneous along all of the fitted values, hsCRP was log (to base e) transformed; the exponentiated coefficients represent the percent change from the geometric mean (anti-log). There were no zero values for hsCRP concentration. We verified the assumptions for a linear model with graphical and statistical tests of the associations between ultra-processed food intake and hsCRP concentration as well as between the fitted values and residuals. We further added the exposure value squared in the model to assess whether there was curvature in the association between ultra-processed food intake and hsCRP concentration. We used locally weighted scatterplot smoothing (LOWESS) models to examine whether there was a threshold for the association between ultra-processed food intake and hsCRP concentration. We also tested with graphical and statistical tests the normality of residuals and homoscedasticity (homogeneity of variance). We used the variance inflation factor to assess collinearity between the potential confounders included in our models. Lastly, to allow for comparison with the three previously published ultra-processed inflammation studies [21,22,23], and given that the assumptions associated with a linear model were not violated, we used linear models. The estimated relative change in hsCRP concentration (mg/L) for each energy-adjusted 100 g increase in ultra-processed food consumption was thus calculated along with robust standard 95% confidence intervals (95%CIs) and *p*-values. We also estimated the variance explained (Pseudo-R^2^) in hsCRP by ultra-processed food consumption via the Cragg-Uhler method [42].

Four different sequential models were fitted: energy-adjusted ultra-processed food as the exposure variable and otherwise unadjusted (model 1), additionally adjusted for sociodemographic characteristics (model 2), and a fully adjusted model that further controlled for lifestyle- and health-related behaviours (model 3). Since previous studies have highlighted body mass index as a potential mediator in the association between ultra-processed food consumption and inflammation [7,21], supplementary analyses were performed by additionally adjusting for body mass index (model 4). It is important to highlight here that we were interested in assessing the “total effect” of ultra-processed food consumption on hsCRP concentration. As such, and given that body mass index was a prespecified mediator, we assessed its possible impact as part of our supplementary analyses (model 4). However, given the cross-sectional nature of our study, we refrained from referring to and did not formally test for mediation [43]. Because body mass index did not qualify as a confounder (see [44]), model 3 was considered the main model. Other studies have also reported differences between sexes in the association between ultra-processed food consumption and inflammation [21,22]. We thus undertook further supplementary analyses to (a) stratify by sex and (b) assess the potential effect modification of sex with ultra-processed food consumption. To explore sex as a possible effect modifier, we added interaction terms between sex and ultra-processed food consumption into the main effects model.

To ensure the sampling methods did not affect the results, sensitivity analyses were performed across all models with the following exclusions: (1) people with hsCRP > 10 mg/L (*n* = 122), which may indicate acute inflammation [45], although these values can also be seen in cases of chronic inflammation [46]; and (2) cases defined by cardiovascular disease mortality (*n* = 567). We conducted further sensitivity analyses on our main model 3 by excluding people with history of non-communicable diseases, such as hypertension (*n* = 555), stroke (*n* = 44), heart attack (*n* = 129), cancer (*n* = 166), diabetes mellitus (*n* = 100), and body mass index ≥30 (*n* = 520).

Lastly, we conducted post hoc analyses by fitting a logistic regression on our main model 3 (adjusted for sociodemographic characteristics and lifestyle- and health-related behaviours) to assess associations between each energy-adjusted 100 g increase in ultra-processed food consumption and the odds of hsCRP at or above 3 mg/L, which is considered a risk factor for cardiovascular events [45].

The analyses were undertaken using R version 3.6.3 (R Development Core Team, Vienna, Austria) [47].

## 3. Results

The current study included 1261 men and 757 women. Table 1 shows the sociodemographic and lifestyle characteristics of participants. The mean age of participants was 57 years. Most people reported that they were married or in a de facto relationship (75.6%) as well as reporting their country of birth as Australia or New Zealand (64.2%). Approximately one quarter of participants were in the top quintile of SEFIA (least disadvantaged; 25.1%) and reported that they had either some study towards or had completed a tertiary degree or diploma (24.0%). Less than a fifth of participants reported that they were current smokers (14.2%) and over one fifth (21.7%) reported that they had engaged in high physical activity over the last six months. Most participants (40.8%) had an average alcohol intake of less than 19 (g/day). The mean body mass index for all participants was 27.8 (kg/m^2^), and the mean proportion of ultra-processed food in the overall diet by weight and energy was 26% (g/day) and 40% (kJ/day), respectively. In terms of ultra-processed food intake in absolute weight and energy, the median was 364.4 (g/day) and 2975.1 (kJ/day), respectively. The median hsCRP concentration for participants was 1.6 (mg/L).

Table 2 details the results of the multivariable adjusted models. In model 1, every 100 g increase in ultra-processed food intake was associated with a 3.6% increase in hsCRP concentration (95%CIs: 1.7–5.5%, *p* < 0.001). After accounting for sociodemographic characteristics and lifestyle- and health-related behaviours in the main multivariable analysis (model 3), the association remained robust (expected relative change in hsCRP: 4.0%; 95%CIs: 2.1–5.9%, *p* < 0.001). The supplementary analyses including all participants and further adjustment for body mass index are also shown in Table 2 (model 4). Part of the association between ultra-processed food intake and hsCRP concentration was independent of body mass index, where every 100 g increase in ultra-processed food intake was associated with a 2.5% increase in hsCRP concentration (95%CIs: 0.8–4.3%, *p* = 0.004). Results remained relatively stable in our sensitivity analyses that excluded people with hsCRP concentrations above 10 mg/L, cardiovascular disease mortality and history of cardiovascular diseases, cancer, diabetes mellitus, and body mass index ≥30 (see Appendix A Appendix A).

There was no evidence of sex interactions (all *p*-values > 0.05 and estimates of interaction range: 0.0–2.6%). The supplementary analyses stratified by sex are shown in Appendix A Appendix A. After accounting for potential confounders in our main model 3, every 100 g increase in ultra-processed food intake was associated with an increase in hsCRP concentration in both men (estimated relative change in hsCRP: 3.5%; 95%CIs: 1.3–5.7%, *p* = 0.002) and women (estimated relative change in hsCRP: 5.5%; 95%CIs: 0.5–10.5%, *p* = 0.032). However, after further adjustment for body mass index (model 4), the association remained robust in men only (estimated relative change in hsCRP for men: 2.8%; 95%CIs: 0.7–4.9%, *p* = 0.010 versus women: 2.4%, 95%CIs: −2.1–6.8%, *p* = 0.296). Post hoc analyses on our main model 3 showed that each 100 g increase in ultra-processed food consumption was associated with 1.08-fold increased odds of hsCRP concentration at or above 3 mg/L after adjusting for sociodemographic characteristics and lifestyle- and health-related behaviours (odds ratio: 1.080; 95%CIs: 1.034–1.128, *p* < 0.001).

## 4. Discussion

This study aimed to examine whether greater ultra-processed food intake was associated with higher hsCRP concentration in a sample of Australian adults. We found evidence of this association, and at least part of this association was independent of body mass index.

Three epidemiological studies have previously tested associations between ultra-processed food consumption and biomarkers of inflammation [21,22,23]. Overall associations with men and women combined were not tested in two of these ultra-processed food-inflammation studies [21,22]. This makes it challenging to compare these studies’ results with the main results from our study. However, our results are partly consistent with another that assessed overall associations in male and female adolescents aged from 17 to 18 years [23]. That study demonstrated direct cross-sectional associations between the consumption of ultra-processed food and concentration of the inflammatory cytokine, interleukin-8 [23]. It also showed that participants with the highest intake of ultra-processed food had increased concentrations of leptin and C-reactive protein compared to participants with the lowest intake, but these associations were less certain given the 95% confidence intervals that crossed zero in both the unadjusted and fully adjusted linear models [23]. These less certain findings, particularly regarding C-reactive protein, may be partly explained by the included sample of adolescents who were exclusively from public schools in a lower socio-economic region of Brazil [23]. The authors of that study noted that these sociodemographic characteristics have been associated with lower consumption of ultra-processed food, with generalisability issues and underestimated effect estimates remaining possible [23]. Indeed, ultra-processed food contributed 26% to total daily energy intake in that Brazilian sample of adolescents compared to 40% in our sample.

While sex was not a significant effect modifier in our study, we conducted sex-stratified supplementary analyses to allow for comparison with previous literature [21,22]. One previous study reported direct prospective associations between the intake of ultra-processed food and interleukin-6 concentrations across two separate cohorts, with one cohort showing an association in women only and the other showing an association in men only [22]. Adiposity did not appear to explain these cohort- and sex-specific findings [22]. Results for the men-only analysis in our study support the data from the second cohort [22], where we also found associations between higher ultra-processed food intake and elevated hsCRP concentration across all models, including additional adjustment for body mass index.

In contrast, for the women in our study, observed associations were not independent of body mass index. These findings are somewhat concordant with another previous ultra-processed food-inflammation study [21], which reported direct cross-sectional associations between ultra-processed food intake and high-sensitivity C-reactive protein in women only and that appeared to be explained by body mass index [21]. These findings suggest that in women, adiposity is a possible intermediate on the causal pathway from ultra-processed food consumption to inflammation. This notion may be explained by the greater accumulation of adiposity, on average, in women compared to men; associations between body mass index and C-reactive protein concentration are suggested to be stronger for women than men [48]. However, cross-sectional studies do not allow for formal tests of mediation [43], and this notion requires further investigation in prospective analyses. However, it is important to reiterate that testing for effect modification by sex in our study showed no evidence of interaction. Given the underrepresentation of women compared to men in our study (37.5%), it is possible that we may not have had adequate power to detect this interaction. Further investigation with more appropriately designed studies is needed.

Our study is consistent with recent systematic reviews and meta-analyses [8,9,10] showing direct associations between intake of ultra-processed food and the prevalence and incidence of common chronic non-communicable diseases, morbidity, and mortality, all of which include inflammation as part of their pathophysiology [49]. Our results are also consistent with a recent systematic review of observational studies and broader whole of diet or dietary pattern analyses [50]. Not unlike the NOVA food classification system, dietary patterns expand beyond isolated nutrients and account for the fact that foods are consumed in complex combinations [50]. This systematic review reported that indices and scores used to assess the inflammatory potential of diets (e.g., Dietary Inflammatory Index) were directly and cross-sectionally associated with inflammatory biomarkers, including C-reactive protein, interleukin-6, tumour necrosis factor-α, and fibrinogen [50]. Pro-inflammatory dietary patterns were characterised by, for example, excess consumption of kilojoule-dense Western-style foods, including red and processed meats, sweets, desserts, fried foods, and refined grains [51]. Similarly, diet scores measuring adherence to healthy or Mediterranean-style diets—rich in fruits, vegetables, fatty fish, poultry, extra virgin olive oil, and whole grains—appeared to be inversely associated with inflammatory biomarkers in cross-sectional analyses [50]. In terms of experimental evidence, our results are also consistent with an earlier meta-analysis of intervention studies showing that Mediterranean diets higher in unprocessed or minimally processed foods were anti-inflammatory [52].

A potential role of the gut microbiota in the link between ultra-processed food intake and inflammation was hypothesised [53]. Preliminary theory posits that extensive food processing leading to the degradation of cell walls within food and acellular compounds (i.e., deconstruction of the food matrix and nutrients not contained within cells, respectively [54]) may impact abnormal absorption and signalling from the gastrointestinal tract as well as its interactions with gastrointestinal microbiota [53,54,55]. Both may in turn promote microbe encroachment on the gastrointestinal wall and a cascade of inflammatory processes [53]. Though not extensively demonstrated in humans, several pre-clinical rodent studies have indicated an effect of advanced glycation end-products (AGEs) formed during the thermal treatment of food products [56,57] and artificial additives common to ultra-processed food (e.g., carboxymethylcellulose [58,59], polysorbate-80 [59,60], saccharin [61], and sucralose [62]) on the gut microbiota composition and activity together with host physiology including pro-inflammatory states. One recent randomised controlled-feeding study in humans reported a detrimental effect of the emulsifier carboxymethylcellulose on the gut microbiota and metabolome, with the authors of that study surmising that carboxymethylcellulose may be contributing to an array of chronic inflammatory diseases [63]. This emerging evidence is certainly suggestive and warrants further investigation to determine the precise features of ultra-processed food that elicit their unhealthful effects.

### 4.1. Limitations and Future Research

Our results should be interpreted with consideration of the following limitations. First, the possible temporality of these associations cannot be established from this single cross-sectional study, and residual confounding cannot be excluded. However, our results were unchanged after adjusting for common confounders and after various sensitivity analyses, which showed that the associations remained relatively stable with or without the inclusion of people with markedly elevated hsCRP concentration. Sensitivity analyses also highlighted that our sampling methods may not have biased results given the stability of effect estimates no matter whether we included cases defined by cardiovascular disease mortality (which occurred after data and sample collection) or participants with a history of non-communicable diseases, such as cardiovascular diseases, cancer, diabetes mellitus, and body mass index ≥30 at baseline.

Second, although the FFQ was not specifically designed to identify ultra-processed food, there is some evidence in certain populations (e.g., New Zealand children [64] and adults from Italy [65] and Mexico [66]) that FFQs have acceptable validity to assess food consumption based on the NOVA food classification system. FFQs have also been reported as the most frequently used dietary data collection tool in reviews investigating ultra-processed food–chronic disease relationships [8]. Nonetheless, some degree of misclassification bias may exist.

Lastly, while the FFQ dietary data used to investigate ultra-processed food intake in the current study were captured over 20 years ago, this may not limit the generalisability of our results since both the participant characteristics and level of ultra-processed food consumption reported in the current study are comparable to current estimates globally. As specified in Table 1, ultra-processed food contributed 40% of total energy intake, which is consistent with the most recent analysis of ultra-processed food intake in a nationally representative sample of Australians taken from the National Nutrition and Physical Activity Survey (2011–2012) [28], where ultra-processed food contributed 42% of total energy. This is also comparable with other estimates from Western countries such as Canada (42%), the United Kingdom (54%), and the United States (56%) [8].

### 4.2. Implications

Historically, and as previously noted, nutrition research has focused on the effect of dietary intakes of energy and macro- and micro-nutrients on human physiology and health, including inflammatory processes together with the mechanistic link between inflammation and chronic diseases. The relevance and novelty of the NOVA food classification system becomes prominent, however, when considering emerging evidence for differential health outcomes that depend on the extent and level of food processing [67,68,69,70]. Indeed, NOVA largely ignores the nutrient profiles of ultra-processed food, instead focusing on the extent and purpose of food processing [11]. One tightly controlled randomised trial in humans that specifically applied the NOVA food classification system in its design [7] demonstrated a causal effect of an ultra-processed versus unprocessed diet on increased energy intake as well as adiposity (both of which have been associated with pro-inflammatory states [71]). While this landmark study targeted different metabolic outcomes, it also showed a within-group reduction from baseline to endpoint in hsCRP concentration when participants were allocated to the unprocessed diet [7]. It also underscored the futility of focusing only on nutrient composition given that the two diets were matched for presented energy, sugar, fat, fibre, and macronutrients [7].

Given the association between ultra-processed food intake and morbidity and mortality [8], there were recent calls urging countries to adopt policy interventions that limit the production, distribution, and dietary intake of ultra-processed food [72]. The importance of our study includes its potential generalisability to other Anglo-European populations and ability to inform and encourage future research investigating the possible biological mechanisms of action involved in the observed associations between consumption of ultra-processed food, chronic non-communicable diseases, and all-cause mortality.

## 5. Conclusions

The current study showed a cross-sectional association between higher ultra-processed food intake and elevated hsCRP as a biomarker of inflammation. Part of the association between consumption of ultra-processed food and hsCRP was independent of body mass index. Further prospective and experimental studies in humans are needed to examine whether this association is causal. Such information will be key to appropriate health messages in the future.

## Figures and Tables

**Figure 1 nutrients-14-03309-f001:**
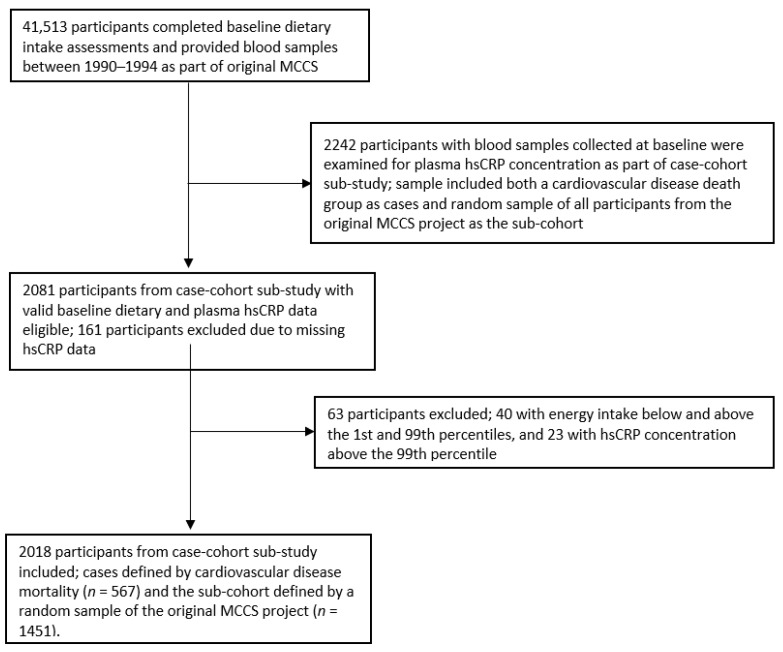
Flow-chart of participant selection. MCCS—Melbourne Collaborative Cohort Study, hsCRP—high-sensitivity C-reactive protein.

**Table 1 nutrients-14-03309-t001:** Descriptive characteristics of the study population.

*n*	Total = 2018
*n* for cardiovascular death cases	Total = 632
*n* for sub-cohort (random sample of all MCCS participants)	Total = 1386
Age (years)—mean (SD)	57.0 (8.8)
Women	757 (37.5%)
Married/de facto	1431 (75.6%)
(In)complete tertiary degree or diploma ^a^	485 (24.0%)
Top quintile of SEIFA ^b^ index (Q5: least disadvantaged)	504 (25.1%)
Born in Australia/New Zealand	1296 (64.2%)
Current smoker	287 (14.2%)
High physical activity score ^c^ (≥6)	438 (21.7%)
Alcohol intake of up to 19 g/day	801 (40.8%)
Body mass index (kg/m^2^)—mean (SD)	27.8 (4.7)
Proportion (%) of ultra-processed food (g/day)—mean (SD)	26.0 (11.4)
Proportion (%) of ultra-processed food (kJ/day)—mean (SD)	40.0 (13.0)
Total ultra-processed food (g/day)—median (interquartile range: Q1, Q3)	364 (248, 518)
Total ultra-processed food (kJ/day)—median (interquartile range: Q1, Q3)	2975 (2091, 4244)
hsCRP concentration (mg/L)—median (interquartile range: Q1, Q3)	1.6 (0.8, 3.6)

^a^ (In)complete tertiary degree or diploma referred to participants who had some study towards a tertiary degree or diploma as well as participants who had completed a tertiary degree or diploma. ^b^ SEIFA Socio-Economic Indexes for Areas [37]. ^c^ Ordinal score based on frequency of walking plus frequency of less vigorous activity plus twice the frequency of vigorous activity, and ranging from 0 to 16 [32,34]. MCCS—Melbourne Collaborative Cohort Study, hsCRP—high-sensitivity C-reactive protein, SD—standard deviation.

**Table 2 nutrients-14-03309-t002:** Cross-sectional associations between ultra-processed food intake and hsCRP concentration (MCCS, 1990–1994).

Main Analyses				
Variable	*n*	Estimated Relative Change in hsCRP Concentration (mg/L) for Each Energy-Adjusted 100 (g) Increase in Ultra-Processed Food Intake (95%CIs)	*p*-Value	R^2^
Model 1 ^a^	2018	3.6% (1.7–5.5%)	<0.001	6%
Model 2 ^b^	1899	4.2% (2.3–6.0%)	<0.001	11.3%
*Model 3 ^c^	1852	4.0% (2.1–5.9%)	<0.001	15.1%
**Model 4 ^d^	1850	2.5% (0.8–4.3%)	0.004	27.7%

Regressions performed with hsCRP on a logarithmic scale. ^a^ Model 1 = unadjusted. ^b^ Model 2 = additionally adjusted for sociodemographic characteristics: sex (men and women), age (continuous), education ((in)complete tertiary degree or diploma, completed high/technical school, (in)complete high/technical school, completed primary school, and (in)complete primary school), country of birth (Australia/New Zealand/Other, United Kingdom/Malta, Italy, and Greece), marital status (married, de facto, divorced, separated, and widow), and SEIFA quintiles (Q1–Q5). Change to *n* due missing values for confounders marital status and SEIFA quintiles. ^c^ *Model 3 = main model additionally adjusted for lifestyle- and health-related behaviours: smoking status (never smoked, current smoker, and former smoker), physical activity over the last 6 months (0 (none), >0 and <4 (low), ≥4 and <6 (moderate), and ≥6 (high)), and alcohol intake (g/day) (lifetime abstainers, ex-drinkers, and up to 19, 20–29, 30–39, and 40+). Change to *n* due missing values for confounder alcohol intake. ^d^ **Model 4 = supplementary analyses additionally adjusted for body mass index (kg/m^2^). Change to *n* due missing values for confounders alcohol intake and body mass index. SEIFA—Socio-Economic Indexes for Areas, 95%CIs—95% confidence intervals.

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
