# Peer review of "Higher Ultra-Processed Food Consumption Is Associated with Greater High-Sensitivity C-Reactive Protein Concentration in Adults: Cross-Sectional Results from the Melbourne Collaborative Cohort Study"

_nutrients, 2022, doi:10.3390/nu14163309_

Round 1

Reviewer 1 Report

The authors examined the association between ultra-processed food intake and CRP based on a well established population based study. The findings are interesting and fill in the knowledge gap in the field.  I have some concerns on the analyses.

1.       Table 2, is it based on log transformed CRP as the dependent variable in multivariable models? Please describe it in the method section. How was zero value of CRP treated in the analysis?

2.       What was the variation of CRP explained by ultra-processed food?

3.       Is there any association between ultra-processed food and elevated CRP (e.g. CRP>3)? It may be more relevant clinically to look at the association between ultra-processed food and elevated CRP.

4.       What is the distribution of the sample by the intake of ultra-processed food?

5.       The authors assumed the association between ultra-processed food intake and CRP to be linear. Did the authors test the non-linear association? Is there a threshold for the association?

Reviewer 2 Report

Manuscript ID nutrients-1842047

Title

Higher ultra-processed food consumption is associated with greater high-sensitivity C-reactive protein concentration in adults: cross-sectional results from the Melbourne Collaborative Cohort Study

The manuscript has relevant and important content for the area of ​​food and nutrition. The text is well written.

I have only one observation, the model adjusted for BMI was significant only for men. It is known that men have more fat-free mass than women and that they are generally more involved in exercise practice. Can it be assessed whether these factors led to the results of model 4?

Results page 8

“However, after further adjustment for body mass index (model 4), the association remained robust in men only (estimated relative change in hsCRP for men: 2.8%; 95%CIs: 0.7% – 4.9%, p=0.010 versus women: 2.4%, 95%CIs: -2.1% – 6.8%, p=0.296)”.

Reviewer 3 Report

This is an interesting study, but there are severe methodological limitations that need to be addressed. The NOVA classification is not unambiguous, and as the authors themselves state, the FFQ used has severe limitations. Several foods could be in NOVA 3 or 4 depending on the actual food consumed - and it is unlikely that this will be picked up. 

The authors need to address this - there appears to be a lot of certainty in the results and the authors make fairly bold claims about associations with inflammation. The intrinsic limitations of the methodology means that results are suggestive and not as definitive as claimed and the authors should address this uncertainty throughout the paper, not tucked away in the "limitations" section.

As minor comment: it would be good to use non-linear models.

Round 2

Reviewer 3 Report

The authors have addressed all but one of much comments. Unfortunately there was a misunderstand regarding linear models: it would be good if the authors could use a transformation - such as splines or polynomes - to explore non-linear relationships.

Author Response

To Dr. Monica Dinu and Dr. Daniela Martini as Guest Editors of the Special Issue “Ultra-Processed Foods, Diet Quality and Human Health”,

Thank you for providing Reviewer 3’s feedback regarding our revised paper and that it would be good if we could use a transformation - such as splines or polynomes - to explore non-linear relationships. However, we are concerned about the scope and aims of our paper as well as a multiple comparisons problem given that we have already conducted a vast number of main, transformatory, supplementary, sensitivity and post hoc analyses that were part of our originally registered analysis plan as well as the reviewers’ previous feedback. Given this was a minor suggestion from Reviewer 3 and that our assumptions for a linear regression were met, these non-linear analyses would be purely exploratory and do not add to previous UPF-inflammation studies that have not conducted such analyses; indeed, the current version of our paper won’t interfere with the interpretation of existing literature since previous UPF-inflammation studies have not explored non-linear relationships. In general, results from non-linear analyses are also more difficult to interpret (see DOI: 10.1002/sim.1638). We kindly ask that the Editors consider this response as addressing Reviewer 3’s feedback. However, if the Editors request these further analyses, we would kindly ask for an extended resubmission timeline as they will require substantial time to incorporate into the paper.

Many thanks,

Melissa and team